# Network-based Active Inference and its Application in Robotics

## Abstract

This paper introduces Network-based Active Inference (NetAIF), a novel robotic framework that enables real-time learning and adaptability in dynamic, unstructured environments. NetAIF leverages random attractor dynamics and the Free Energy Principle (FEP) to simplify trajectory generation through network-topology-driven attractors that induce controlled instabilities and probabilistic sampling cycles. This approach allows robots to efficiently adapt to changing conditions without requiring extensive pre-training or pre-calculated trajectories. By integrating learning and control mechanisms within a compact model architecture, NetAIF facilitates seamless task execution, such as target tracking and valve manipulation. Extensive simulations and real-world experiments demonstrate NetAIF's capability to perform rapid and precise real-time adjustments, highlighting its suitability for applications requiring high adaptability and efficient control, such as robotics tasks in the energy and manufacturing sectors.

## 1 Introduction

### 1.1 Overcoming Automation Challenges with Advanced Learning Methods

The World Energy Employment 2023 report by the IEA highlights a significant shift towards clean energy jobs, which now surpass fossil fuel employment, driven by a 40% rise in clean energy investment over the past two years. Despite economic and geopolitical challenges, the energy sector has seen growth in employment, particularly in solar PV, wind, EVs, and battery manufacturing. However, a shortage of skilled labor remains a key challenge, underscoring the need for targeted training and policy support to develop a workforce suited for the energy transition (IEA, 2023).

In response to these labor challenges, automation is playing an increasingly critical role in advancing the clean energy sector. Robotics, in particular, offers a promising solution to enhance operational efficiency and safety. However, to maximize the potential of robotics in complex and dynamic environments, sophisticated learning methods are required. One such approach, Deep Reinforcement Learning (DRL), has emerged as a leading candidate for enabling autonomous robotic systems in tasks like control, manipulation, and decision-making. Yet, despite its potential, DRL faces notable barriers to widespread adoption in the energy sector.

### 1.2 Deep Reinforcement Learning (DRL)

DRL combines the decision-making power of reinforcement learning (RL) with the pattern recognition capabilities of deep learning (DL). This allows robots to learn and adapt through trial and error, improving performance over time. DRL is increasingly explored for enabling autonomy in control and manipulation tasks in real-world environments by training agents to recognize complex patterns in data and make informed decisions.

However, DRL requires large amounts of data and time for agent training, as well as expert-designed reward functions to guide learning. Creating these reward functions demands substantial knowledge and engineering resources, as they must accurately capture desired outcomes, agent actions, and constraints. Poorly defined reward functions can lead to suboptimal or unsafe behavior (Sutton & Barto, 2020). Thus, while powerful, DRL may not always be the most practical or cost-effective approach for every application.

### 1.3 ACTIVE INFERENCE AS A NEXT-GENERATION LEARNING METHOD

Active Inference (AIF) is an advanced framework from neuroscience that is now being applied in robotics to help agents minimize *surprisal*—the unexpectedness of observations—without relying on traditional reward-based approaches like deep reinforcement learning (DRL). The goal of the agent is to reduce surprisal, mathematically expressed as $-\log p(o)$, where $p(o)$ represents the probability of an observation $o$.

Since directly minimizing surprisal is often impractical, the agent minimizes *variational free energy* $\mathcal{F}$, which serves as an upper bound on surprisal:

$$\mathcal{F} = \mathbb{E}_{q(s_t)}\left[\log q(s_t) - \log p(o_t, s_t)\right] \geq -\log p(o_t)$$

In this expression, $q(s_t)$ is the approximate posterior over states $s_t$, and $p(o_t, s_t)$ is the joint likelihood of observing $o_t$ given the state $s_t$. By minimizing $\mathcal{F}$, the agent balances *accuracy* (matching observations) and *complexity* (keeping the model simple), continuously refining its predictions and actions to reduce prediction error.

While AIF holds significant promise for creating adaptive robotic systems, its real-world deployment faces challenges due to the complexity of model design and high computational demands (Lanillos et al., 2021). Nonetheless, its potential to enhance flexibility, durability, and adaptability makes it a powerful alternative to traditional DRL techniques

### 1.4 NETWORK-BASED ACTIVE INFERENCE (NETAIF)

To overcome the limitations of both DRL and traditional AIF approaches, we propose Network-based Active Inference (NetAIF), a novel framework that leverages network dynamics to simplify trajectory calculations and enhance efficiency. Rooted in key AIF principles such as entropy and surprise minimization, NetAIF builds on the Free Energy Principle (FEP), which posits that systems self-organize by minimizing surprisal or prediction error. By harnessing the inherent dynamics of a network, NetAIF computes trajectories more efficiently than traditional AIF methods, reducing the need for complex mathematical models while enabling agents to adapt to dynamic environments in real-time. This streamlined approach makes NetAIF highly suitable for real-world robotic applications, offering significant improvements in both speed and computational cost.

## 2 NETWORK-BASED ACTIVE INFERENCE

### 2.1 NOTABLE CHARACTERISTICS

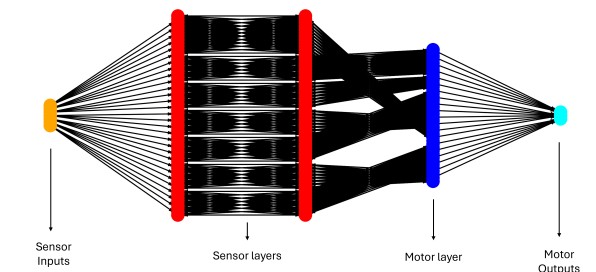

Figure 1: NetAIF network diagram for target-tracking task: parameters that determine the network structure such as number of layers, strides were determined through hyper parameter search

NetAIF introduces explicit feedback loops between hidden layers, deliberately inducing controlled instabilities. Through extensive simulations and real-world experiments, we observe that these feedback mechanisms enable the network to explore the state space more thoroughly, leading to improved adaptability in dynamic environments. This behavior is evidenced by the robot's ability

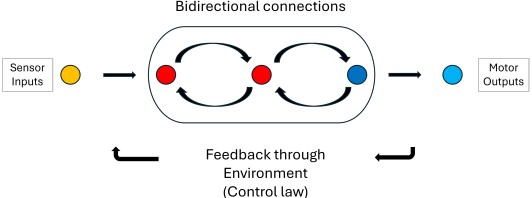

Figure 2: bidirectional connection in hidden layers: the schematic diagram shows how the instability is induced within the hidden layer and how such instability is controlled via the external control law through feedback

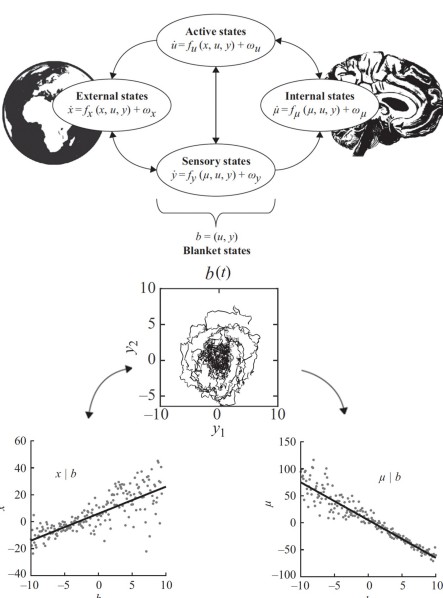

Figure 3: AIF brain and world - External states (world) are mirrored by internal states (brain). The active and sensory states (blanket states) couple external to internal states-rendering the system open. The (far from equilibrium steady-state) dynamics of each state is described with stochastic differential equations ($w$ is a stochastic fluctuation). The images were adapted and modified from Parr et al. (2022)

to rapidly adjust to changing targets without pre-training or pre-calculated trajectories (Brown, 2021)(Refer to Figs. 1 and 2). Unlike Recurrent Neural Networks (RNNs), where feedback is implicit (Mienye et al., 2024), NetAIF actively manipulates network dynamics to push the system into unstable regions. These feedback loops enhance oscillatory patterns, similar to neuron firing sequences, that persist even after training. This random bursts of node activity can be observed in the supplementary video, further highlighting the parallels with brain function. The introduction of these instabilities enables the system to maintain dynamic behaviors, known as itinerant (wandering) dynamics (Kaneko & Tsuda, 2003; Friston & Ao, 2012), allowing it to continuously adapt to changing environments.

This intentional instability serves two purposes. First, it reflects autovitiation, where self-induced instability maintains dynamic behavior in Active Inference (AIF) systems (Friston & Ao, 2012). Second, it supports a continuous cycle of hypothesis testing, akin to Bayesian inference, where the system anticipates and adjusts based on discrepancies between expected and actual sensory data.

Operating within the AIF framework, NetAIF interacts with its environment through blanket states—sensory states gather information, while active states influence the environment, maintaining

a Non-Equilibrium Steady State (NESS). This dynamic feedback loop ensures the system remains stable yet flexible, minimizing prediction errors in real time.

By balancing sensory inputs and active states, NetAIF continuously refines its internal model, optimizing performance in complex, uncertain environments, much like Bayesian inference, allowing for real-time adaptation and trajectory optimization.

NetAIF also replaces traditional activation functions with a discrete weight-assigning mechanism, designed to reset node weights and maintain NESS. By leveraging the constant interaction between sensory and active states, NetAIF remains in a state of continuous exploration, avoiding local minima and ensuring that it adapts dynamically to new challenges. This stochastic function enhances the network's ability to explore different states, preventing it from being trapped in local optima.

Additionally, NetAIF integrates learning and control, guiding motor outputs with clear task-specific control laws. These laws break tasks down into sub-goals, such as aligning objects, allowing even non-experts to define behaviors without deep control theory knowledge. For instance, in a valve manipulation task, control instructions guide the network to minimize errors by aligning the vector of the valve's position with the one of the end effector. This ensures precise orientation and movement, making the system more intuitive and effective for real-world applications. This user-friendly approach facilitates seamless integration of learning and control.

---

**Algorithm 1** Main loop of the NetAIF model

---

1: **Initialize** all model parameters and weights
2: **while** system is running **do**
3:     Prediction_Error = $Desired\_State - Current\_State$
4:     Input_signals = $Prediction\_Error$
5:     **for** each weight $w$ in all weights **do**
6:         **if** magnitude of associated signal > threshold **then**
7:             Set $w = new\_weight\_value()$
8:         **end if**
9:     **end for**
10:    Input_to_hidden = $Input\_signals \times W\_input\_hidden$
11:    Feedback = $Hidden\_signals\_prev \times W\_hidden\_hidden$
12:    Hidden_signals = $Input\_to\_hidden + Feedback$
13:    Hidden_signals_prev = $Hidden\_signals$
14:    Outputs = $Hidden\_signals \times W\_hidden\_output$
15:    Motor_Commands = $Outputs$
16:    Send motor commands to actuators
17: **end while**

---

The core of the NetAIF framework is outlined in Algorithm 1. Each cycle calculates the prediction error between current and desired states, which updates network weights dynamically. If a signal exceeds a set threshold, its weight is reset to ensure stability. Feedback loops in the hidden layers facilitate adaptive behavior and robust trajectory generation. Motor commands are derived from the hidden layers and sent to the actuators, enabling real-time adjustments. This continuous feedback allows NetAIF to quickly adapt to changing environments, making it ideal for dynamic tasks like PV panel inspection.

## 2.2 THE RANDOM ATTRACTOR

To represent the NESS behavior in NetAIF, Random Dynamical Systems (RDS) are employed, providing a framework to understand complex systems driven by stochastic processes. In particular, random pullback attractors (Caraballo & Han, 2016), also known as stochastic basins of attraction, describe how NetAIF's state evolves over time in response to environmental uncertainty. Expressed as $\varphi(t, \omega, x)$, where $t$ is time, $\omega$ represents randomness, and $x$ is the state variable, these attractors characterize regions in the state space where the system tends to settle. The random attractor $\mathcal{A}(\omega)$ pulls trajectories towards it, ensuring that NetAIF remains adaptive and stable within its NESS framework, despite external randomness.

This is formalized by:

$$\lim_{t \to \infty} \text{dist}\left(\varphi(t, \theta_{-t}\omega, B), \mathcal{A}(\omega)\right) = 0$$

where $\varphi(t, \theta_{-t}\omega, B)$ represents the state of the system at time $t$, $\theta_{-t}\omega$ is the time-shifted random noise, where $\theta$ is a shift operator that moves the noise backward in time by $t$ units. This term captures

the idea that the noise affecting the system at time $t$ is related to the noise that occurred in the past. $B$ is a bounded set of initial conditions, and $\text{dist}(X, Y)$ denotes the distance between sets $X$ and $Y$.

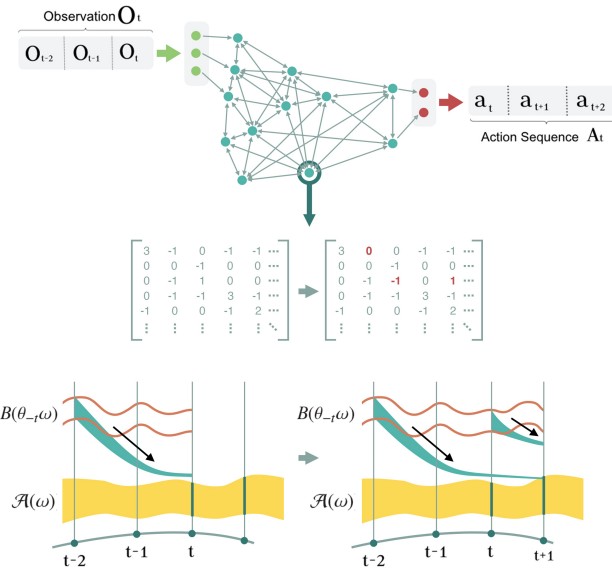

Figure 4: Abstract representation of a random pullback attractor, $\mathcal{A}$, and the random set, $B$. While the weights of the network are updated randomly (shown in matrix format), a flow from the random set emerges and gets attracted to the attractor.

This convergence process can be understood as a stochastic diffusion in parameter space, driven by increasing the amplitude of random fluctuations on parameters (e.g., connection weights) in regions of high free energy. As the system approaches free energy minima, these random fluctuations are attenuated, resulting in a more stable and precise arm trajectory. Such system dynamics can be described by a stochastic differential equation (SDE) in the form of a Langevin equation (Karl, 2019):

$$dx = -\nabla F(x)\, dt + \sqrt{2\Gamma}\, dW$$

where $x$ represents the system's parameters, $F(x)$ is the free energy landscape, $\Gamma$ is the diffusion coefficient, and $W$ is a Wiener process. This equation captures the interplay between the deterministic drift towards free energy minima and the stochastic exploration of the parameter space, which ultimately shapes the arm's trajectory.

It is worth noting that the optimization process in NetAIF is inherently local because free energy is an extensive quantity, meaning that the system's total free energy is the sum of the free energies of its individual components. The variational free energy, which approximates the true free energy, is calculated using local prediction errors. Some predictions are clamped with high precision, fixed, or strongly influenced by the desired outcomes, defining the attracting set, which represents the desired sensor inputs or the target state of the system. Minimizing variational free energy by reducing local prediction errors guides the network model towards the attracting set.

This local optimization process enables the system to efficiently navigate the free energy landscape without requiring global computations or information propagation across the entire network. By iteratively updating its local components based on prediction errors and external control laws, the system converges towards the desired states.

The roots of this learning scheme can be traced back to early formulations of self-organization in cybernetics (Ashby, 1947) (Ashby, 1956) and are connected to stochastic thermodynamics (Ao, 2008) (Seifert, 2012). These connections highlight the consistency of the design principle with the fundamental concepts underlying the FEP. This principle drives the network model to minimize prediction errors, guiding the entire network towards a stable regime, resulting in smooth and efficient arm movements.

## 3 REAL-WORLD VALIDATION AND PERFORMANCE EVALUATION

We conducted three key experiments using the Lite6 6-DoF robotic arm from UFactory, operating at 100 Hz: a pose-matching task, a target-tracking task, and a valve-turning task. Each experiment was designed to evaluate different aspects of the NetAIF framework, including its real-time trajectory generation and adaptability in dynamic environments.

### 3.1 POSE-MATCHING TASK

In the pose-matching task, which served as a benchmark, the desired joint pose was directly fed into the system. The NetAIF model calculated waypoints using attractor dynamics to generate smooth and efficient trajectories, guiding the robot to the specified pose without explicit path planning algorithms. The control law simply aimed to match the current joint position with the desired one. As a result, the Lite6 arm smoothly reached the target position, showcasing the effectiveness of NetAIF for trajectory generation.

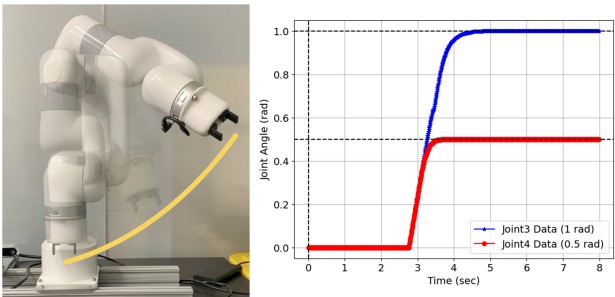

Figure 5: Network Output Signal for Pose Matching Task

### 3.2 TARGET-TRACKING TASK

In the target-tracking task, the robotic arm followed an AprilTag detected by a RealSense D455 camera, with tracking accuracy enhanced by a Kalman filter (Kam et al., 2018). Reference vectors were used to align the robot's roll, pitch, and yaw with the moving target. Notably, the arm tracked the marker in real time without pre-training, demonstrating NetAIF's capability for adaptive and flexible motion planning in dynamic environments.

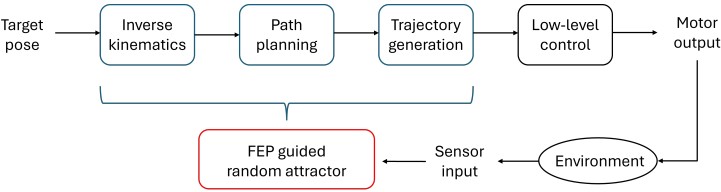

Figure 6: Motion planning process

### 3.3 VALVE-TURNING TASK

For the valve-turning task, the Lite6 arm was used to manipulate valves of different shapes (triangle, square, circle) while the Intel RealSense D455 camera provided valve localization. This task further demonstrated NetAIF's real-time adaptability. The swift and efficient performance of the NetAIF model can be attributed to its FEP-guided path generation, combined with random attractor dynamics. As illustrated in Fig. 6, these random attractor dynamics replace conventional motion planning components. Unlike some of the traditional methods, where the entire trajectory is pre-calculated

or trained, NetAIF generates the trajectory in real-time by continuously feeding sensor data to the random attractor, allowing for more flexible and adaptive motion planning.

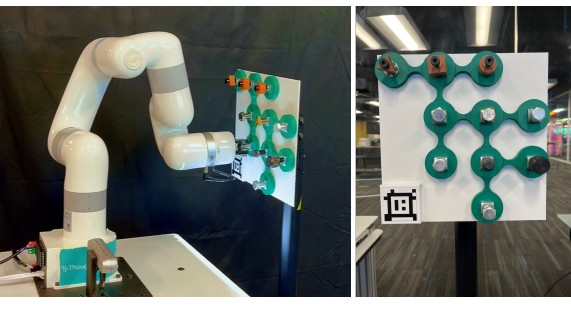

Figure 7: Valve-turning experiment setup. Left: The Lite6 robotic arm manipulates valves of various shapes. Right: Examples of valve shapes and bolts used in the experiments.

### 3.4 EFFICIENT DEPLOYMENT AND ADAPTABILITY OF NETAIF: PERFORMANCE METRICS AND FLEXIBILITY

Table 1 presents the performance metrics for the NetAIF model, evaluated on an 8-core Intel Core i9 (I9-9880H) 2.4 GHz processor without GPU support. The network's update cycle ranged from approximately 5ms to 7ms, as detailed in Table 2, resulting in remarkably short training times—around 7 seconds for the valve-turning task (evaluated using the Lite6 robot from UFactory as shown in Fig. 7) and about 8 seconds for the target tracking task. Once the network is trained, the trajectory values become smoother with relatively small random fluctuations. This smoothness reflects the efficiency of the network's attractor dynamics, which generate real-time adjustments based on sensor data, allowing for precise tracking without requiring pre-calculated trajectories.

What sets NetAIF apart is its computational efficiency and rapid adaptability. Designed for swift deployment with minimal overhead, NetAIF efficiently utilizes stored weight values and attractor dynamics to reduce the computational footprint, making it highly suitable for resource-constrained systems. Unlike traditional neural networks that require extensive retraining or significant computational resources when applied to new tasks or environments, NetAIF facilitates quick adaptation to different robotic platforms and tasks without substantial retraining. This minimizes deployment overhead and allows for seamless transitions between tasks and environments, enhancing operational flexibility in ways that standard neural networks may not readily support.

Table 1: NetAIF Model Metrics

| Metric | Pose-Matching | Target-Tracking | Valve-Turning |
|---|---|---|---|
| Network Size (No. of Nodes) | 132 | 176 | 332 |
| Network Size (No. of Connections) | 1212 | 1616 | 1872 |
| Network Size (No. of Bytes) | 10224 | 13632 | 16304 |
| No. of Iterations to Convergence | 955 | 1230 | 1413 |

### 3.5 TIME-LAGGED CROSS-CORRELATION ANALYSIS

Fig. 8 shows a cross-correlation analysis between a marker's position in the X, Y, and Z directions and six robot joints, revealing how different joints influence the marker's movements over time. The analysis highlights coordinated robot motion driven by the network's attractor dynamics. Joints 2 and 5 exhibit delayed correlations with the marker's X position, indicating their role in larger, slower movements after other joints have initiated motion. In contrast, joint 1 shows a stronger, immediate influence on the marker's Y direction, reflecting its control over base-level adjustments. Z-axis motion involves more complex interactions, with joints 2 and 3 leading, suggesting their importance in vertical positioning. These leading and lagging behaviors reflect the robot's kinematics, where

base joints initiate broader movements and distal joints fine-tune or stabilize them, enabling precise and coordinated control.

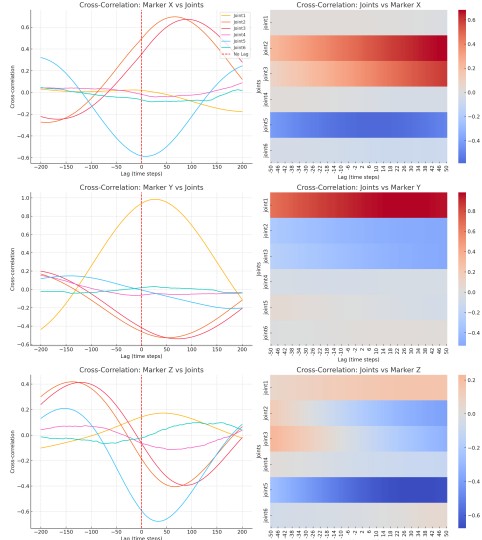

Figure 8: Time-lagged cross-correlations

## 3.6 MOTION PLANNING AND PERFORMANCE SUMMARY

The total motion planning time for both the target-tracking and valve-turning tasks, involving real-time visual processing, is summarized in Table2 and Fig. 9. For the target-tracking task, the NetAIF model achieves an average planning time of 6.7 milliseconds, highlighting its ability to operate efficiently in environments requiring frequent replanning due to dynamic changes and moving targets. Despite a standard deviation of 16.16 milliseconds, which reflects variability due to factors such as fluctuating frame rates and environmental dynamics, the model consistently delivers fast, responsive performance with a median time of 5.23 milliseconds. This balance of speed and adaptability makes the system well-suited for real-time applications.

Table 2: Summary of time taken to generate values by the network

| Statistic | Target-tracking (ms) | Valve-turning (ms) |
|---|---|---|
| Mean time | 6.7 | 4.53 |
| Standard deviation | 16.16 | 2.09 |
| Median time (50th percentile) | 5.23 | 5.38 |
| 25th percentile | 4.56 | 2.75 |
| 75th percentile | 5.80 | 6.21 |

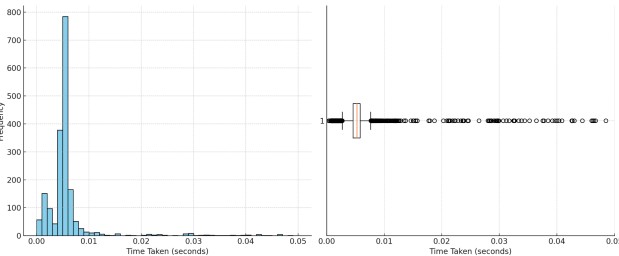

Figure 9: Total motion planning time for target-tracking task

In comparison, the valve-turning task exhibits even greater efficiency, with an average planning time of 4.53 milliseconds and a much lower standard deviation of 2.09 milliseconds, indicating more consistent behavior. The median time of 5.38 milliseconds is close to that of the target-tracking task, but the tighter spread of the data (as seen in the 25th and 75th percentiles) suggests that the valve-turning task benefits from a more predictable environment, resulting in reduced variability in planning time.

When compared to other state-of-the-art algorithms, the performance of the NetAIF model stands out. Traditional methods such as PRM and Hybrid RRT-PRM can take up to 482 milliseconds to generate plans under similar conditions, due to the significant computational overhead involved in path updates (Jermyn, 2021). Similarly, UAV-based systems that rely on visual processing report planning times ranging from 50 to 500 milliseconds in dynamic environments (Cui et al., 2022). Even with the higher variability in target-tracking, the NetAIF model's mean planning time of 6.7 milliseconds far surpasses these algorithms, making it an exceptional solution for real-time, dynamic tasks that require frequent replanning without sacrificing speed or responsiveness.

## 4 DISCUSSION

### 4.1 RANDOM PULLBACK ATTRACTOR—EMPIRICAL EVIDENCE IN NETAIF

Building on the concept of a random pullback attractor previously discussed, our observations of the NetAIF model provide strong empirical evidence supporting its presence within the network's dynamics. Despite stochastic fluctuations and varying initial conditions, the network consistently converges toward a stable region in its state space over time. This behavior reinforces the idea that an underlying attractor governs the system's long-term trajectory, aligning with the theoretical framework of random pullback attractors in Random Dynamical Systems (RDS) theory (Caraballo & Han, 2016).

### 4.1.1 EVIDENCE FROM CONVERGENCE PATTERNS AND OPTIMIZATION PRINCIPLES:

Figure 10 illustrates that the time required for the network to reach equilibrium increases linearly with the number of nodes, even though the network's complexity grows nonlinearly as more nodes are added. In these simulations, we employed fully connected networks without environmental disturbances to isolate the effect of network size on convergence time. The observed linear relationship across different network sizes suggests that the network dynamics are governed by an attractor that scales predictably with the network's architecture.

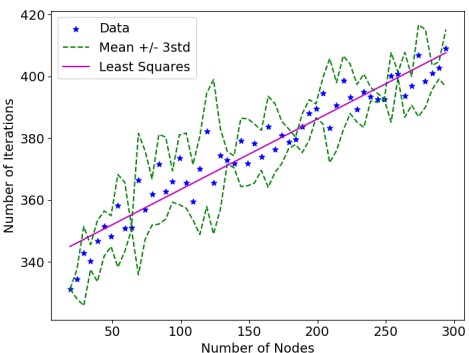

Figure 10: Iterations to Equilibrium - In this simulation, the network was *fully* connected without environment disturbances to see how the complexity of the network affects the convergence time. 5% window size was used to seek local outliers. The plot shows the data are following a linear trend without any outliers

This consistent convergence pattern implies that, as the network evolves, it effectively minimizes a potential function—similar to the minimization of free energy in the Free Energy Principle (FEP). Moreover, this natural tendency aligns with the Least Action Principle (LAP) in classical mechanics

(Wang, 2006), which states that a system evolves along the path of least action, minimizing the integral of the Lagrangian over time. Essentially, systems tend to follow the most efficient trajectory between two states.

In the context of NetAIF, the network dynamics appear to inherently seek the most efficient temporal path toward stabilization, regardless of initial conditions. This suggests that the network is optimizing its behavior by minimizing a functional analogous to action, thereby aligning with universal optimization principles found in physics. Such alignment underscores the robustness and efficiency of NetAIF's attractor dynamics, contributing to its ability to adapt and stabilize effectively in dynamic environments.

### 4.1.2 CONSISTENCY ACROSS DIFFERENT RUNS:

Further evidence comes from observing that networks with identical structures but different initial weight values and stochastic fluctuations converge to similar behaviors. Figure 11 compares the weight values of identical connections between different simulation runs. Despite variations in individual weights due to random initializations and updates, the overall network behavior remains consistent across runs. This robustness indicates that the attractor dynamics are primarily determined by the network's topology rather than specific parameter values.

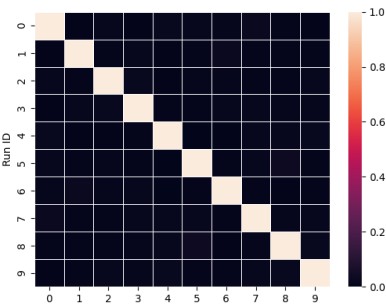

Figure 11: Comparison of Weight Values of Identical Edges between Runs with Analogous Behaviors

This phenomenon mirrors the concept of degeneracy in biological systems, where different components or pathways produce similar functions or behaviors. In neuroscience, for example, diverse neural circuits can give rise to the same functional output due to the brain's highly interconnected and redundant architecture (Edelman & Gally, 2001). Similarly, in genomics, different genetic sequences can result in the same phenotypic trait due to alternative genetic pathways.

The NetAIF model's ability to converge to similar behaviors despite differences in weights reflects this principle of biotic self-organization. The network's topology acts as a blueprint that shapes its functional dynamics, much like how the structure of biological systems determines their emergent properties. This connection to biological concepts underscores the naturalness and potential robustness of NetAIF's design.

## 5 CONCLUSIONS

The Network-based Active Inference (NetAIF) model presents an efficient approach to real-time adaptive intelligence in robotics, utilizing random attractor dynamics and the Free Energy Principle (FEP) to enable rapid adaptation to unpredictable environments without requiring extensive pre-training or high computational resources. Its real-time feedback ensures precise control and flexible adaptation, making it ideal for industries like energy, where cost-efficiency and adaptability are crucial. Unlike Deep Reinforcement Learning (DRL), which demands significant training and computational power, NetAIF offers a more streamlined, cost-effective solution for tasks such as inspections and maintenance. For a comparison with DRL methods, see the companion paper (Anonymous, 2024).

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
