# OpenReview forum: "Network-based Active Inference and its Application in Robotics"
_ICLR.cc/2025/Conference — Submitted to ICLR 2025_

### Official Review · Reviewer_KT7F · 2024-11-01

**Soundness:** 1
**Presentation:** 1
**Contribution:** 1
**Rating:** 1
**Confidence:** 4

**Summary:**

This paper proposes an active-inference inspired neural network architecture that can be used for various tasks in robotics applications. It is claimed that the neural network architecture follows several biological and physical principles, such as the minimization of free energy, or the principle of least action. Several experiments show that the model can be used for pose matching, target tracking and a valve-turning task with a real 6-DOF robotic arm.

**Strengths:**

The idea to use active-inference or related biologically-inspired principles for robotics is interesting, and indeed, such methods can be much more sample-efficient than a standard application of Deep RL, which often tends to be sample-inefficient and have long-training cycles.

**Weaknesses:**

However, the paper is not written well. I was often puzzled throughout reading the paper, as the proposed model and architecture's properties are described without really introducing a method. The methodology seems very unclear, and the link to the literature is mostly not discussed, so that I was unable to form a clear idea of what the method is and what the contribution (with respect to the related work in the literature) is. Moreover in the experiments there were no comparisons with other methods, even though the method is claimed to improve significantly over deep RL baselines. Besides that, I was at least expecting a comparison with a more control-theoretic baseline, such as a (well-tuned) PD controller.

===== POST-REBUTTAL EDIT =====

I thank the authors for their rebuttal. I also removed the offending sentence, it was not meant as a personal/subjective comment, however I understand that it can be taken as such, for that I apologize. Overall my grade remains the same, and I strongly recommend the authors to introduce their method with clear mathematical notation. All the reviewers had trouble understanding the methodology, although it is agreed that Active Inference is a promising venue for robotics.

A few comments below on the rebuttal:

"The explanation of controlled instability (Section 2) is thoroughly detailed, describing the explicit bidirectional connections in the hidden layer and the external feedback loop that stabilizes the network. " However all the reviewers including me had trouble understanding the controlled instabilities introduced by the method, and the weight resetting.

"NetAIF is a novel framework that introduces a fundamentally new approach to trajectory generation and control, leveraging stochastic attractor dynamics and active inference principles. Due to its novelty, it has limited direct relevance to existing methods in the literature, which may contribute to a perceived gap in its contextualization. It is important to note that NetAIF’s approach represents a significant departure from traditional methodologies, making comparisons less straightforward." We're unable to appreciate perhaps the novelty of the approach because of (i) lack of clear understanding of the methodology, (ii) lack of clear comparisons, (iii) lack of detailed explanations of the algorithm and the implementation. I suggest the authors to significantly revise the introduction, perhaps then the contributions will be clearer.

**Questions:**

I will list here some comments and questions that I had while reading the paper:

Section 1
- much more citations are needed in section 1. We're missing the connection to the literature.
- in what way is active inference 'advanced'?
- why is minimizing surprisal often impractical?
- mention the relation/links to variational inference (Bayes) in section 1.3
- "By minimizing F, the agent balances accuracy (matching observations) and complexity (keeping the model simple)"
This is not clear from the text. Characteristics of the method are often explained without really introducing the method, e.g.,
"NetAIF computes trajectories more efficiently than traditional AIF methods" How is this done?
Section 2
- It is not clear what Figure 1 is, is it a fully connected network, or a convolutional neural network? or something else (RNN, etc.)?
- "NetAIF introduces explicit feedback loops between hidden layers, deliberately inducing controlled instabilities. Through extensive simulations and real-world experiments, we observe that these feedback mechanisms enable the network to explore the state space more thoroughly, leading to improved adaptability in dynamic environments."
What does it mean a controlled instability? what do you mean by 'thoroughly'? These claims are not supported by the experiments. I strongly recommend the authors to start from clear mathematical definitions.
- Figure 2 is not clear. What does red signify? what does blue signify for instance?
- "NetAIF actively manipulates network dynamics to push the system into unstable regions." How is this done? It does not seem to be discussed in the experiments adequately. And in general, instability is a huge problem in robotics. Or perhaps the authors mean something else? (see my comment above also, these points would be clarified by starting from mathematical definitions)
- "These feedback loops enhance oscillatory patterns, similar to neuron firing sequences, that persist even after training. This random bursts of node activity can be observed in the supplementary video, further highlighting the parallels with brain function." Again this seemed very vague to me (hence the comparison to a manual, rather than a clear scientific paper).
- Figure 3: "rendering the system open" What does open mean?
- What is a non-equilibrium steady state? Not clear from the text, is there a relation to Lyapunov stability?
- Algorithm 1 is not clear and not explained, not is it self-contained. How is desired state determined? What is new_weight? The algorithm seems to be missing a loop, given that there is feedback. Nor is the feedforward mapping of inputs to outputs properly introduced (unless there is only one hidden layer).
- What is the Free Energy in Algorithm 1? How is it playing a role?
- "By iteratively updating its local components based on prediction errors and external control laws, the system converges towards the desired states." -> How is this to be proved?

Section 3 (Experiments)
- How would you deal with constraints?
- How do you get the desired joint poise? Inverse Kinematics?
- There are no comparisons. The task seems especially easy, so I would expect a well-tuned PD controller to do equally well here.
section 3.4
- How is the network trained?

Section 4 (Discussions)
- "Moreover, this natural tendency aligns with the Least Action Principle (LAP) in classical mechanics" -> how?
- "The network dynamics appear to inherently seek the most efficient temporal path toward stabilization, regardless of initial conditions. This suggests that the network is optimizing its behavior by minimizing a functional analogous to action, thereby aligning with universal
optimization principles found in physics. " Appear and suggest are not rigorous enough, especially given that you're proposing a network structure whose novelty hinges on this explanation.

---

> ### Author Response · Authors · 2024-11-26
>
> We appreciate the time and effort Reviewer KT7F has invested in providing feedback. However, we would like to raise some concerns regarding the appropriateness and relevance of parts of the review. Specifically:
> 1.	Misunderstanding of Key Concepts:
> o	Several points raised by the reviewer appear to reflect a misunderstanding of material already presented in the paper. For example: The explanation of controlled instability (Section 2) is thoroughly detailed, describing the explicit bidirectional connections in the hidden layer and the external feedback loop that stabilizes the network.
> 2.	Focus on Unrelated Comparisons:
> o	The reviewer has suggested comparisons to methods such as PD controllers or baselines like PRMs and Hybrid RRT-PRMs. While these may be valid for some studies, NetAIF’s focus is on trajectory generation through biologically inspired network dynamics. It is not directly comparable to conventional trajectory planners or simple feedback controllers. We believe this focus was made clear in the paper, but we will further emphasize it in the revised manuscript.
> 3.	Review Tone and Scope:
> o	The review describes the paper as reading “more like a manual than a scientific paper” and expresses dissatisfaction with the link to the literature. While we welcome constructive criticism, such comments are subjective and not directly tied to the paper’s scientific contributions. This language could be perceived as dismissive, and we request a more objective engagement with the material.
>
> Below you will find the brief response:
> Comment: methodology is unclear, and the connection to the literature is weak.
> Response:
> NetAIF is a novel framework that introduces a fundamentally new approach to trajectory generation and control, leveraging stochastic attractor dynamics and active inference principles. Due to its novelty, it has limited direct relevance to existing methods in the literature, which may contribute to a perceived gap in its contextualization. It is important to note that NetAIF’s approach represents a significant departure from traditional methodologies, making comparisons less straightforward.
>
> Comment: There is a lack of clarity on how active inference is advanced in this work and why minimizing surprisal is impractical. The link to variational free energy and Bayesian principles is weak.
> Response:
> 1.	Advancements in Active Inference: NetAIF builds on the active inference framework by introducing stochastic attractor dynamics for trajectory generation. This novel approach enables real-time adaptability without requiring extensive pre-training or pre-calculated trajectories, as is common in other methods.
> 2.	Use of Variational Free Energy and Bayesian Principles: Variational free energy and Bayesian principles were briefly discussed in the paper to introduce the broader context of active inference (AIF) but are not directly utilized in NetAIF itself. Instead, NetAIF focuses on biologically inspired network dynamics to adapt to dynamic environments.
> 3.	Why Minimizing Surprisal is Impractical: Minimizing surprisal directly requires full knowledge of the system’s dynamics and observations, which is infeasible in real-world tasks. By minimizing variational free energy, NetAIF provides an efficient approximation that balances accuracy and simplicity. This trade-off will be clarified in the revised manuscript.
>
> Comment: How are constraints handled? How are joint positions determined? The control approach is unclear, and a well-tuned PD controller might perform equally well.
> Response:
> 1.	Handling Constraints: Constraints are managed through task-specific control laws and the inherent limits of the robotic hardware (e.g., joint limits and velocity constraints). These ensure safe and bounded operation during task execution and will be described explicitly in the revised manuscript.
> 2.	Joint Position Determination: NetAIF computes joint positions in real-time by minimizing prediction errors based on sensory feedback. These positions are then executed by a low-level joint position controller provided by the robot hardware.
> 3.	Comparison with PD Controllers is inappropriate: While PD controllers can handle simple tasks, NetAIF is designed for scenarios with dynamic noise and non-linearities, where PD controllers require extensive manual tuning and lack adaptability. For example, in the AprilTag tracking task, NetAIF demonstrated real-time adaptability to marker variability and noise, which would be infeasible for a standard PD controller.
>
> Comment: Algorithm 1 lacks clarity. What is meant by weight resetting
> Response:
> 1.	Algorithm Explanation: Algorithm 1 describes a threshold-based weight resetting mechanism (mentioned in the paper) that dynamically adjusts weights to maintain stability and adaptability. When the magnitude of a signal exceeds a predefined threshold, weights are recalibrated to prevent the network from diverging or getting stuck in local minima.

---

### Official Review · Reviewer_NKHz · 2024-11-03

**Soundness:** 2
**Presentation:** 1
**Contribution:** 2
**Rating:** 3
**Confidence:** 3

**Summary:**

This paper proposes a novel framework for robust learning of real-world robotics tasks in unstructured environments. The framework, called NetAIF, simplifies trajectory generation by relying on stochastic Random Dynamical Systems (RDS) and training model weights using the Free Energy Principle. Explicit feedback loops are introduced between hidden layers to control instabilities and enable real-time learning of three robotic tracking and interaction tasks.

**Strengths:**

- Interesting idea of using stable attractor dynamics for trajectory generation.
- Accompanying videos of tracking experiments suggest reasonable tracking performance.

**Weaknesses:**

- Metrics and baselines are unclear. The only metric presented is average planning time in Section 3.6. However, no concrete comparisons with existing approaches like PRMs and Hybrid RRT-PRM as mentioned in the same section. Given the simplicity of the tasks evaluated, the paper also does not discuss why a simple PID tracking controller would not work, at least for the first 2 tasks.
- The tasks presented lack significant details to allow for a fair evaluation of the presented approach. For instance, the action space used in each of the tasks is not clear. Similarly, the type of robot controller used is also unclear and could have a significant bearing on the results of the experiment. Is the gripper state part of the system state?
- Sensory information here seems to be a direct measurement of the state. In general, this is only true for simple tasks. For instance, imagine a driverless car. Sensory information will generally be obtained through sensors like cameras without an explicit finite-sized state space. It is not clear how this approach would apply to a more general scenario where the relationship between sensory information and state information is not clearly defined.
- Paper lacks clarity on the data used for training. How are the weights trained for each task? Is Algorithm 1 used to train the network weights? If this is the case, how do the authors ensure that training on the robot like this is safe?

**Questions:**

Clarifications:
- What does weight resetting mean? How does resetting work and how does this affect the dynamics of the network?
- How do you validate performance in the presence of environmental disturbance? All the tasks investigated here seem to have little to no environmental disturbance.

---

> ### Author Response · Authors · 2024-11-26
>
> 1. Metrics and Baselines
> Comment: Metrics and baselines are unclear. No comparisons with PRMs or Hybrid RRT-PRM are provided.
> Response: The primary objective of this paper is to showcase NetAIF’s unique performance characteristics rather than provide an exhaustive benchmarking study. Table 1 highlights the compactness of the network (small size and quick convergence) and demonstrates its efficiency in stabilizing behavior across tasks. Table 2 complements this by showing the mean motion planning times, which emphasize NetAIF’s ability to generate real-time trajectories. While PRMs or Hybrid RRT-PRM comparisons are valuable, such an analysis would require a dedicated study and is outside the scope of this work. The reference to motion planning times was included to give a sense of NetAIF’s speed and efficiency. This intent will be clarified in the manuscript to avoid any misunderstanding. We will also better explain the implications of the metrics provided to reinforce the significance of NetAIF’s contributions.
>
> 2. PID Tracking Controller
> Comment: Why would a simple PID tracking controller not work for at least the first two tasks?
> Response: PID controllers are effective in simple, predictable environments but struggle with dynamic noise and non-linearities. NetAIF's stochastic attractor dynamics allow real-time adaptation without pre-calculated trajectories, enabling robust performance even in tasks like AprilTag tracking, which involve marker variability and noise. This would be infeasible for PID controllers, which require manual tuning and lack adaptability.
>
> 3. Action Space and Controller
> Comment: Details about the action space, robot controller, and gripper integration are unclear.
> Response:
> Action Space: The action space corresponds to joint positions of the robotic arm. NetAIF computes desired joint positions based on sensory feedback, which are sent to the controller.
> Robot Controller: The tasks use a low-level joint position controller provided by the robotic hardware, ensuring precise execution of NetAIF’s high-level trajectories.
> Gripper Integration: In the valve-turning task, the gripper state is part of the system state, dynamically updated via sensory feedback to ensure precise manipulation. This integration ensures seamless coordination between the arm and gripper.
>
> 4. Data for Training and Weight Adjustment
> Comment: How are weights trained? How is safety ensured?
> Response:
> Training Process: NetAIF operates without the need for pre-collected datasets or traditional offline training. Instead, the weights are dynamically adjusted during task execution using a threshold-based resetting mechanism, as detailed in Algorithm 1. When the magnitude of an associated signal exceeds a predefined threshold, the corresponding weights are recalibrated to maintain system stability and adapt to changes in real-time. This lightweight, online adaptation approach ensures NetAIF is well-suited for dynamic environments without the overhead of pre-training.
> Safety Considerations: NetAIF is inherently a control-integrated framework designed with task-specific control laws to ensure safe and stable operation. The system operates within predefined joint limits and velocity constraints of the robotic hardware. Furthermore, the prediction-error-driven feedback mechanism helps prevent erratic behavior by ensuring consistent trajectory updates. All experiments were conducted in controlled environments to validate safety. For real-world deployments, additional safeguards such as collision detection and virtual boundaries can be incorporated. These points will be expanded in the manuscript to address any ambiguity around safety.
>
> 5. Environmental Disturbance
> Comment: Tasks seem to have little to no environmental disturbance.
> Response:
> Dynamic Disturbances in Target-Tracking: The target-tracking task included camera jitter, marker movement variability, and frame rate fluctuations. Despite these challenges, NetAIF dynamically adapted trajectories without pre-training, as reflected in the standard deviation of planning times in Table 2.
> Stochastic Attractor Dynamics: NetAIF’s stochastic attractor dynamics inherently address environmental uncertainties by introducing controlled instabilities that stabilize trajectories under noise.
> Disturbances in Other Tasks: The valve-turning task involved real-world challenges like uneven valve shapes and bolt misalignments, while the pose-matching task served as a baseline for trajectory efficiency. Future work will test NetAIF in more explicitly noisy environments.

---

### Official Review · Reviewer_UKSm · 2024-11-04

**Soundness:** 3
**Presentation:** 3
**Contribution:** 4
**Rating:** 6
**Confidence:** 4

**Summary:**

This paper introduces Network-based Active Inference (NetAIF), a robotic framework that leverages active inference principles and network dynamics to enable robots to adapt in real-time to dynamic environments. Built upon the Free Energy Principle (FEP), NetAIF supposedly simplifies trajectory generation using network-driven attractors, which introduce controlled instabilities and probabilistic sampling. These design choices supposedly allow for efficient, rapid responses to environmental changes without extensive pre-training, contrasting with deep reinforcement learning (DRL) approaches that require complex reward structures.

**Strengths:**

I think active inference principles with network dynamics is a nice idea specifcially given that application of active inferecne has been a challenge without cumbersome approximations. So looks like your approach of integrating active inference with network dynamics is promising, potentially offering a fresh perspective in robotic control.

It is also interesting idea to diverge form optimization at every level and think of exploration and perturbations to converge towards good solutions.

The mode seems to be reduce computational overhead, making it suitable for resource-constrained cases like robots.

The authors provide thorough experimental validation, showcasing the framework's effectiveness across various tasks.

**Weaknesses:**

I do have some concerns regarding the work.

Lack of Structure in Descriptions: The initial sections of the paper, particularly the introduction, lack clear organization, making it difficult to follow the progression of ideas. A section on DRL is presented with no experiments on DRL later on in the paper.  The authors could reintroduce the problem of challenges of control more generally and issues with active inference that you are trying to address here.  start with the general challenges in robotic control, then introducing active inference and its limitations, followed by how NetAIF addresses these issues. Next would be related works and then a proper formulation of your methodology. This would provide a clearer progression of ideas for the readers to follow.

Relevant Literature : Also I dont feel the related works section is done well.  It doesn't cover earlier works that are compuationaly cheap and related such as PMP (https://link.springer.com/article/10.1007/s10514-016-9563-3) or other adaptive/neural control methods. Basically, refer to methods that are learable, adaptive and computationaly cheap to provide a relevant review and comparison.

Comparative Analysis: Then  paper does not sufficiently compare NetAIF against established approaches in terms of accuracy and efficacy, limiting the context of its contributions. I undersand there is another work by the authors submitted to ICLR 2025. I suggest to merge the works given the same underlying idea. I believe a combined paper would show a promising work with applications and comparisons making it stronger or more impactful.

Convergence Guarantees: The method’s ability to ensure convergence to high accuracy while incorporating constant perturbations is not adequately addressed, raising concerns about stability and reliability. You could provide theoretical guarantees or empirical evidence demonstrating the system's stability under various perturbation conditions.

Relevance to ICLR: The focus on robotic control, as opposed to learning and representation, may limit its appeal to broader ICLR community although this is not a limitation of the work per se.

**Questions:**

Comparative Metrics: What are the specific performance metrics used to evaluate NetAIF against existing methods, and how do these metrics support the claims of improved adaptability and efficiency?

Accuracy and Stability: How does the framework guarantee convergence to a high accuracy given the introduction of controlled instabilities and random perturbations? What mechanisms are in place to handle potential divergence?

Long-Term and Large system Performance: What measures are taken to evaluate the long-term performance and adaptability of NetAIF in highly variable environments or highly redundant robots as perturbations may make the system unstable.

---

> ### Author Response · Authors · 2024-11-26
>
> We sincerely thank Reviewer UKSm for their thoughtful and constructive feedback. Below, we address each point in detail:
> ________________________________________
> Strengths
> We appreciate the reviewer’s recognition of NetAIF’s novelty, computational efficiency, and robustness, as well as its integration of active inference principles with network dynamics. These points align with the primary contributions of our work, and we are pleased to see them highlighted.
> ________________________________________
> Weaknesses
> 1. Lack of Structure in Descriptions
> Comment: The initial sections lack clear organization, making it difficult to follow the progression of ideas. There is no section on DRL to provide context.
> Response: We will reorganize the introduction to clearly present challenges in robotic control and active inference, followed by a description of how NetAIF addresses these challenges.
> ________________________________________
> 2. Relevant Literature
> Comment: The related works section is incomplete.
> Response: Due to space limitations, the related works section was abbreviated to focus on NetAIF’s methodology. In the revised manuscript, we will expand this section to:
> •	Discuss similar control schemes that use controlled instabilities.
> •	Include a biological example of explicit bidirectional control and inherent instability characteristics, which inspired NetAIF’s design.
> ________________________________________
> 3. Comparative Analysis
> Comment: The paper lacks comparisons with other approaches in terms of accuracy and efficacy.
> Response: This paper is focused on introducing the novel NetAIF methodology. Comparative analyses with traditional approaches, including DRL-based controllers, are presented in a separate paper, which is cited in this work. We will ensure this reference is clear and highlight key distinctions, such as NetAIF’s computational efficiency and adaptability, to provide additional context.
> ________________________________________
> 4. Convergence Guarantees
> Comment: The method’s convergence under perturbations is not adequately addressed.
> Response: NetAIF’s convergence is ensured through controlled instabilities in the hidden layers, which are stabilized by external feedback loops. While formal convergence proofs are outside the scope of this paper, empirical results demonstrate stability under perturbations.
> ________________________________________
> 5. Relevance to ICLR
> Comment: The focus on robotic control may limit the paper’s appeal to ICLR audiences.
> Response: NetAIF introduces biologically inspired network dynamics and active inference principles, offering a novel perspective that bridges learning and control. To enhance its appeal, we will emphasize NetAIF’s adaptive learning capabilities and highlight its real-time weight and trajectory adjustments as part of the broader learning paradigm.
> ________________________________________
> Questions
> 1. Comparative Metrics
> Comment: What are the specific performance metrics used to evaluate NetAIF?
> Response: NetAIF is evaluated based on:
> •	Real-time adaptation: Demonstrated through dynamic trajectory computation without pre-training.
> •	Computational efficiency: Measured via planning time in the results.
> •	Robustness: Assessed through experiments handling dynamic noise and environmental changes.
> ________________________________________
> 2. Accuracy and Stability
> Comment: How does the framework guarantee convergence in the presence of controlled instabilities and random perturbations?
> Response: Stability is achieved through:
> •	Controlled instabilities: Bidirectional connections enable exploration while preventing divergence.
> •	Feedback loops: External mechanisms dynamically stabilize the network by correcting prediction errors.
> Empirical results validate stability under perturbations, and we will expand this discussion in the manuscript.
> ________________________________________
> 3. Long-Term Performance
> Comment: What measures are taken to evaluate long-term adaptability in highly variable environments?
> Response: NetAIF’s feedback-driven architecture ensures continuous adaptation to changing conditions. Experiments, such as AprilTag tracking and valve manipulation, demonstrate its real-time adaptability.

---

### Official Review · Reviewer_Kg1A · 2024-11-07

**Soundness:** 2
**Presentation:** 3
**Contribution:** 3
**Rating:** 3
**Confidence:** 5

**Summary:**

The work at first sight is very interesting as it seems to address some of the challenges in active inference for robotics, reads well, and showing its functioning in a real robot is very much appreciated. It also shows some novelty over previous approaches. However, there is plenty of explanations missing on the methodology used. For instance, where is the learning? Furthermore, results are not deeply analysed. Thus, it is complicated to understand the level of contribution of the approach.

**Strengths:**

* Easy to read
* Novelty of neural network active inference with pullback attractor.
* Interesting behaviour of unstable regions.
* Experiments with real robots

**Weaknesses:**

* The introduction is to broad and to superficial in all three items selected. The energy transition seems to far away from this work. Also only Deep RL is mentioned. What about MPC with learning. In the active inference section survey is the only work reference. But it would be more informative how this work differs from previous robotics works. e.g., An empirical study of active inference on a humanoid robot, A novel adaptive controller for robot manipulators based on active inference, End-to-end pixel-based deep active inference for body perception and action, The Free Energy Principle for Perception and Action: A Deep Learning Perspective, etc.
Include a brief comparison table highlighting key differences between their approach and the cited active inference robotics works. This would give clearer guidance on how to improve the introduction and positioning of their work.

* Methods are unclear. What is the state, what is being minimized and learnt how it is performed. The diffusion process, etc. A lot of further explanation is needed. Provide
1.	A clear definition of the state space
2.	An explicit statement of what quantity is being minimized
3.	A step-by-step description of the learning process
4.	A more detailed explanation of how the diffusion process is implemented in their model
5.	Key methodological details you feel are missing, such as the objective function and learning algorithm equations

* Results analysis can be certainly improved. For instance, pose matching only shows 2-DOF results in joint angles not in the task space. This is not pose. Note that pose considers position and rotation of the end-effector.
Provide full 6-DOF pose results including end-effector position and orientation.

The tracking what is the input output of the NetAIF? Why you need a kalman filter? AIF can do filtering. so the NetAIF is only computing the controller? Improve the flow diagram with notation and input output and explanation.

The valve experiment. "manipulate valves of different shapes (triangle,
square, circle)" this is not shown. It is not clear what is the NetAIF computing and what is the engineering part. Provide analysis of valve manipulation for different shapes and a clear distinction between NetAIF computations and engineering components

Extra remark. Figures should be further explained, just the title is not enough.

**Questions:**

* Where is the learning? Algorithm 1 only shows execution of the net. Provide a separate algorithm or flowchart that explicitly shows the learning process, including weight updates and any optimization steps.
* What is the Free energy landscape (what is the equation? or what is x? how this affect the weights?) how is the diffusion coefficient being computed and  dW.
1.	Provide the explicit equation for the free energy landscape
2.	Clearly define all variables (e.g., x)
3.	Explain how these components relate to the network weights
4.	Detail how the diffusion coefficient is calculated
5.	Explain what dW represents in their implementation and how it is used

* Active inference agents explore when the model is unknown. But here authors express that the system is pushed into unstable regions due to the feedback loops. Provide an explanation of the potential advantages or disadvantages of this method compared to standard active inference exploration techniques.

References:
"For a comparison with DRL methods, see the companion paper (Anonymous, 2024)." This information should be in this paper. And it is marked as under review in ICRA but it can be found as under review in ICLR2025. Better to put all the info in one paper than split into two as it reduces its impact.

The High Road to Active Inference is a chapter. May be better to cite the book and refer to the chapter.

---

> ### Author Response · Authors · 2024-11-27
>
> Weaknesses
> 1. Introduction and Context
> Comment: The introduction is broad and superficial, with unrelated discussion on the energy transition and insufficient comparison to prior active inference robotics work.
> Response:
> We will revise the introduction to focus on challenges in active inference for robotics and NetAIF’s motivation, minimizing unrelated discussions. While the related works section was abbreviated due to space constraints, we will expand it to:
> •	Discuss similar control schemes using controlled instabilities.
> •	Include a biological example of explicit bidirectional control and inherent instability characteristics that inspired NetAIF’s design.
> ________________________________________
> 2. Methodology Clarity
> Comment: Key components like the state space, optimization objectives, and diffusion process are unclear.
> Response:
> The state space in NetAIF is defined by joint positions and orientations. NetAIF does not employ an explicit optimization policy; instead, it relies on attractor dynamics to guide behavior.
> The Langevin equation is included to conceptually illustrate the random attractor phenomenon within the free energy landscape. It highlights how deterministic and stochastic features combine in NetAIF’s framework, rather than serving as a computational element.
> We will:
> •	Explicitly define the state space as joint positions and orientations.
> •	Clarify the purpose of the Langevin equation as a conceptual tool for understanding attractor dynamics.
> ________________________________________
> 3. Results and Analysis
> Comment: The results focus on joint angles and omit critical end-effector position and orientation data.
> Response:
> Our results emphasize the trajectory generation aspect of NetAIF by showing joint angle relationships over time, which is essential to demonstrating its adaptability. End-effector orientation is not relevant to the trajectory generation process for target tracking task. On the other hand, end-effector position data, along with joint angles, is already included in the paper.
> ________________________________________
> 4. Algorithm and Flow Diagram
> Comment: Algorithm 1 lacks details on the learning process, including weight updates. The flow diagram also needs better notation and explanation.
> Response:
> The learning process in NetAIF is achieved through threshold-based weight reset mechanisms, where weights are adjusted dynamically based on prediction errors. These updates enable real-time adaptation and trajectory generation.
> We will:
> •	Improve the flow diagram with clearer notation and a more detailed explanation of its components.
> ________________________________________
> 5. Comparison to Engineering Components
> Comment: The distinction between NetAIF computations and engineering components is unclear.
> Response:
> We will explicitly differentiate NetAIF’s computations (e.g., trajectory generation, adaptation) from the engineering components (e.g., joint controllers, kinematics). For example, NetAIF computes trajectories, while lower-level controllers execute them.
> ________________________________________
> 6. Experimental Validation
> Comment: The valve manipulation experiment lacks details on the engineering and computational aspects.
> Response:
> The valve manipulation experiment setup is demonstrated in the supplementary video. The computational aspects of valve turning are consistent with other tasks, leveraging the same NetAIF framework for trajectory generation and adaptation.

---

> > ### Author Response · Authors · 2024-11-27
> >
> > Questions
> > 1. Learning in Algorithm 1
> > Comment: Algorithm 1 does not show learning, weight updates, or adjustments.
> > Response:
> > Algorithm 1 explicitly includes threshold-based weight reset mechanisms, central to NetAIF’s learning. These mechanisms dynamically adjust weights based on prediction errors exceeding predefined thresholds, enabling real-time adaptation.
> > To address any misunderstanding, we will revise the manuscript to clarify how the threshold-based mechanism represents the learning process in NetAIF.
> > ________________________________________
> > 2. Diffusion Process
> > Comment: How does the diffusion process work, and how is dW computed?
> > Response:
> > The Langevin equation provides a conceptual understanding of random dynamical systems combining deterministic and stochastic phenomena. It illustrates how NetAIF leverages these principles for attractor dynamics within the free energy landscape.
> > The diffusion process and dW are not directly computed but are included to enhance understanding of the framework. We will revise the manuscript to emphasize the conceptual nature of the Langevin equation.
> > ________________________________________
> > 3. Free Energy Landscape
> > Comment: Provide an explanation of the free energy landscape.
> > Response:
> > NetAIF is inspired by active inference principles, particularly the balance of deterministic and stochastic dynamics in attractor frameworks. However, it does not explicitly minimize the free energy landscape. Instead, the free energy landscape is a theoretical framework illustrating how NetAIF integrates these dynamics.
> > We will:
> > •	Clarify that the free energy landscape is used conceptually, not as a computational objective.
> > •	Highlight how active inference principles guide NetAIF’s adaptive mechanisms.
> > ________________________________________
> > 4. Components and Weight Updates
> > Comment: Explain how state variables relate to network weights and how dW is implemented.
> > Response:
> > State variables in NetAIF represent joint positions and orientations, which are dynamically updated through sensory feedback. These interact with network weights via a threshold-based mechanism, adjusting weights when prediction errors exceed predefined thresholds.
> > The term dW, referenced in the Langevin equation, is conceptual, illustrating stochastic components of attractor dynamics. It is not directly implemented but helps explain how random and deterministic features combine to drive adaptive behavior.
> > We will clarify these points in the revised manuscript.
> > ________________________________________
> > 5. Feedback Loops
> > Comment: Explain how feedback loops are implemented and their role.
> > Response:
> > Feedback loops are integral to NetAIF’s learning mechanism. They dynamically correct errors between sensory inputs and motor outputs, enabling the network to adapt to changes. While explicit bidirectional connections in hidden layers introduce controlled instabilities to explore the state space, feedback loops stabilize the network by minimizing these instabilities and guiding trajectories.

---

> > ### Comment · Reviewer_Kg1A · 2024-12-02
> > **Understanding the method is essential when there are no benchmarks**
> >
> > Dear Authors,
> > Thanks so much for your effort to address answer to my questions. While the approach presented seems very promising, it was very difficult to understand the proper methodology. Maybe due to the bias or lack of my background knowledge, but the threshold reset as the mechanism for (learning) adaptation should be clearly explained. What are the dynamics of the neurons, then? how the system stabilizes?, all this information is hindered in the Langevin equation, which is not really implemented in the system. Furthermore, It is a free energy principle approach but it does not seem to minimize the Variational Free Energy. All this explanation confuses the reader.
> >
> > The other option is to reflect the potential of the approach in the results, through benchmarking and statistical evaluation. This is missing in this work. Furthermore, the clarity in input-output is also core to understand what the algorithm is really doing. An example of author's response, which makes difficult to understand the real implementation: "The state space in NetAIF is defined by joint positions and orientations", I do not understand what this actually mean. The joint positions in the 3D space? the joint angle? or the end-effector position and orientation. Then "End-effector orientation is not relevant to the trajectory generation process for target tracking task", This is only in position tracking. You can have pose tracking (as it was mentioned in the text) where you track position and orientation. Anyway, I am not sure what joint orientations means then and what are the trajectories being output and adapted. A mathematical formalization will solve this confusions.
> >
> > Reading other reviewers comments, I really think that the work has a clear potential, but the clarity and maturity is still missing. The amount of work that should be performed at this stage is too huge to not being reviewed again. Thus, I will maintain the score. I understand that this outcome may be not the one that the authors were expecting, but I hope that you get the comments in a constructive manner and improve the paper quality.

---

### Author Response · Authors · 2024-11-27

We respectfully request a re-evaluation of our submission, as the feedback and scores may not fully reflect the paper’s contributions and its central focus.

Through this response, we aim to address points where most confusion arises. Specifically, our paper adopts a holistic simulation-based approach to present NetAIF as a novel framework for robotics, prioritizing practical implementation and experimental validation. The mathematical formulations included in the paper were developed later, primarily to gain insights into the underlying mechanisms and provide theoretical grounding, rather than to derive the framework from scratch.

Key clarifications:

Holistic Approach:

The design and implementation of NetAIF emphasize real-time adaptability and dynamic control, leveraging features like random attractor dynamics and explicit feedback loops. This ensures the framework is practical, efficient, and applicable to dynamic tasks.
The approach prioritizes practical implementation supported by clear procedural descriptions (e.g., Algorithm 1) rather than heavy reliance on mathematical derivations.
Balance Between Theory and Application:

The mathematical equations provide insights into the mechanisms (e.g., Free Energy Principle and random attractors) but are not the foundation of the framework’s development. This balance ensures accessibility while grounding the work in established theoretical principles.
Clarity in Contributions:

NetAIF introduces explicit bidirectional control, dynamic feedback loops, and random attractor dynamics that distinguish it from existing approaches like Deep Active Inference (DAIF). These unique contributions are clearly detailed in Section 2 and supported by empirical evidence in the discussion.
While we recognize the need for improvement in some areas, such as technical clarity and additional details for reproducibility, the extreme scores across all categories seem disproportionate. The paper introduces significant innovations and provides a solid foundation for real-world applications, warranting a more balanced evaluation.

We kindly request a re-evaluation to ensure the paper’s contributions, novelty, and relevance are fairly recognized. Thank you for your time and consideration.

---

### Meta-Review · Area_Chair_bNYR · 2024-12-20

**Metareview:**

This paper introduces the Network-based Active Inference framework (NetAIF) for real-time trajectory generation and adaptation in robotic tasks. While the idea of leveraging biologically inspired active inference principles is intriguing, the reviewers unanimously highlighted significant shortcomings in the paper’s clarity, methodology, and experimental rigor. Despite the authors’ rebuttal, critical issues remain unresolved, necessitating rejection.

The reviewers expressed consistent difficulty in understanding the proposed methodology. Key components, such as the "controlled instability" mechanism and the weight-resetting process, were ambiguously described, with no formal mathematical definitions or detailed explanations provided. The experiments lacked sufficient detail, failing to specify key metrics and offering no meaningful comparisons to established baselines, such as PID controllers or traditional planning methods. Additionally, the paper presented limited real-world applicability, with unclear handling of environmental disturbances and insufficient evidence to support claims of robust adaptability. Given these fundamental issues, the paper does not meet the standard required for acceptance at ICLR.

**Additional Comments On Reviewer Discussion:**

During the rebuttal period, the reviewers raised critical concerns about the paper’s clarity, methodology, and experimental validation. Key issues included the lack of detailed mathematical definitions and explanations for core concepts such as "controlled instability" and the weight-resetting mechanism. Reviewers also noted that the experiments lacked proper metrics, baseline comparisons, and sufficient details about the action space and control methodology, making it difficult to assess the framework's contributions.

The authors’ rebuttal addressed did not manage to cover the reviewers’ concerns. Their responses often reiterated vague descriptions without introducing concrete evidence or clarifications. The lack of new experiments, baselines, or detailed benchmarks cannot be overlooked when assessing this paper.

Weighing the discussion, it was evident that the reviewers remained unsatisfied with the authors’ responses. The many open issues significantly limit the paper's contributions and its relevance to ICLR.

---

### Decision · Program_Chairs · 2025-01-22

Reject